# Synthesize, if you do not have: Effective Synthetic Dataset Creation Strategies for Self-Supervised Opinion Summarization in E-commerce

**Tejpalsingh Siledar♥, Suman Banerjee♣, Amey Patil♣, Sudhanshu Shekhar Singh♣,**
**Muthusamy Chelliah♣, Nikesh Garera♣, Pushpak Bhattacharyya♥**
♥Department of Computer Science and Engineering, IIT Bombay, India
♣Flipkart, India

## Abstract

In e-commerce, opinion summarization is the process of condensing the opinions presented in product reviews. However, the absence of large amounts of supervised datasets presents challenges in generating both aspect-specific and general opinion summaries. Existing approaches have attempted to address these challenges through synthetic dataset creation (SDC). However, general opinion summarization models struggle to generate summaries faithful to the input reviews whereas aspect-specific opinion summarization models are limited due to their reliance on human-specified aspects and seed words. To address this, we propose SDC strategies tailored for general and aspect-specific opinion summarization. We experimented on three e-commerce test sets: Oposum+, Amazon, and Flipkart. For general opinion summarization, pre-trained language model (PLM) fine-tuned on our general synthetic dataset surpass the SOTA on average by **2.3** R1 points. Faithfulness evaluation metrics and human evaluations indicate that our model-generated summaries are more faithful to the input compared to others. For aspect-specific opinion summarization, PLM fine-tuned on our aspect-specific synthetic dataset surpass SOTA by ~ **1** R1 point without the aid of any human-specified aspects or seed words.

## 1 Introduction

In the e-commerce domain, customer reviews are essential for making informed buying decisions. However, going through all the product reviews before making a decision is overwhelming. Opinion summarization is the process of summarizing the opinions presented in the reviews (Hu and Liu, 2006; Wang and Ling, 2016; Angelidis and Lapata, 2018). However, it is infeasible and expensive to obtain reference summaries at a large scale for general and aspect-specific opinion summarization.

| MultimodalSum |
| --- |
| This is a nice teapot, but the color is not as bright as the picture. It is more of a dark turquoise than a light blue. I was hoping it would be more of an aqua blue, but it is more like a dark aqua. It still looks nice, but I would have preferred the color to be more like the photo. |

| Our Model |
| --- |
| This is a beautiful teapot, but the color is not the same as shown in the picture. It is more of a dark turquoise. The color is a little darker than the picture, but it is still a beautiful color. The only thing I don't like about it is that the handle is very hard to open. |

**Table 1: General Opinion Summarization.** Example of *faithfulness issue* in MultimodalSum compared to our model. There is no support for the information in red parts in the corresponding input reviews.

Bražinskas et al. (2020); Amplayo and Lapata (2020) proposed creating synthetic pairs by sampling one of the reviews as a pseudo-summary. However, careful curation of synthetic datasets is essential in ensuring that the models generate summaries that are faithful to the input as shown in Table 1. **In this regard**, we propose a synthetic dataset creation (SDC) approach for general opinion summarization that leverages lexical (R-1 F1 (R1) (Lin, 2004)) and semantic (cosine) similarities for pseudo-summary selection.

Amplayo et al. (2021) proposed constructing synthetic datasets for aspect-specific opinion summarization. They trained a multiple instance learning (MIL) module (Keeler and Rumelhart, 1991) with silver-standard labels obtained using seed words to filter out aspect-containing sentences. Shen et al. (2023) proposed two solutions for generating synthetic datasets without the MIL module. Their NLI-LOO method eliminated the need for seed words but still relied on human-specified aspects. This reliance on human-specified aspects or

**AceSum**

**Ports** - the hdmi ports stopped working and the tv would go black for about three to five seconds every few minutes. it has all of the inputs for us, and more.

**Our Model**

**Ports** - i have had this tv for a few months now and i have had no issues with it, it has all of the hdmi ports i need and the picture quality is great.

**Table 2: Aspect-specific Opinion Summarization.** AceSum vs. Our Model generated summaries for the aspect *Ports*. Our model does not rely on any human-specified aspects and seed words for training, making our approach generalizable to other domains where such aspects and seed words are not defined.

seed words limits aspect-specific opinion summarization to only a few domains where such aspects or seed words are defined. **In contrast**, our proposed solution overcomes this limitation by not relying on any human-specified aspects or seed words and outperforms all existing approaches. Table 2 shows an example aspect-specific summary.

In general opinion summarization, the *input* consists of **reviews**, and the *output* is a **general summary**, whereas, in aspect-specific opinion summarization, the *input* consists of **reviews and aspects**, and the *output* is an **aspect-specific summary**. Our contributions are:

1. For general opinion summarization, a matrix-based synthetic dataset creation strategy that leverages lexical (*R1*) and semantic (*cosine*) similarities for pseudo-summary selection. Pre-trained language model (PLM) fine-tuned on our synthetic dataset of $945K$ instances outperforms the SOTA on average by **2.3** R1 points and generates summaries that are more faithful to the input reviews than alternatives, as evidenced by automatic and human evaluations (Section 2.1, Table 5, 7, & 9).

2. For aspect-specific opinion summarization, an aspect-specific synthetic dataset creation strategy that does not rely on any human-specified aspects or seed words making our approach generalizable to any opinion summarization domain. Pre-trained language model (PLM) fine-tuned on our synthetic dataset of 350K instances outperforms the state-of-the-art by ~ **1** R1 point (Section 2.2, Table 6).

| Pre-defined aspect-seed words mapping | |
|---|---|
| *Pre-defined mapping* | looks: looks color stylish looked pretty
quality: quality material poor broke durable
size: fit fits size big space |
| **Automatic aspect mapping** | |
| *Fine-grained clusters* | selfie camera, camera, camera quality, back camera, ...
screen, display, display, screen quality, ...
battery life, battery backup, battery capacity, battery, ...
processor, intel processor, ... |
| *Coarse-grained clusters* | display, display
camera, camera, camera, ...
battery, battery
processor |
| *Aspect mapping* | display: screen, display, screen quality, ...
camera: selfie camera, camera, camera quality, ...
battery: battery life, battery backup, ...
processor: processor, intel processor, ... |

**Table 3:** An example of the pre-defined aspect-seed words mapping and our automatic aspect mapping.

## 2 Synthetic Dataset Creation (SDC)

### 2.1 General Opinion Summarization

Following Bražinskas et al. (2020), we assume that a review $r_i$ can serve as a summary for a set of reviews $D_i$. This lets us create training points $(D_i, r_i)$, similar to what the model will experience during inference. $D_i$ is limited to size $k$, enabling comparison with previous works where the number of input reviews is fixed (Chu and Liu, 2018).

$$\mathbf{M} = \lambda_1 * \mathbf{M}_{sim} + \lambda_2 * \mathbf{M}_{rouge} \qquad (1)$$

$$\mathbf{M} = \begin{bmatrix} 0 & & & & \\ & 0 & & & \\ & & 0 & & \\ & m_{ij} & & 0 & \\ & & & & 0 \end{bmatrix} \begin{matrix} r_1 \\ r_2 \\ . \\ . \\ r_N \end{matrix} \qquad (2)$$

$$s_i = \text{mean}(\text{topk}(\mathbf{M}[r_i:])) \qquad (3)$$

We encode reviews using sentence-transformers (Reimers and Gurevych, 2019) and construct two matrices $\mathbf{M}_{sim}$ and $\mathbf{M}_{rouge} \in \mathbb{R}^{N \times N}$ by computing cosine similarity and R1 scores between them. We obtain a unified matrix $\mathbf{M}$ by adding the two matrices. A cell $m_{ij}$ of the matrix $\mathbf{M}$ corresponds to a score between reviews $r_i$ and $r_j$. To prevent self-comparisons, we assign zero to the diagonal entries of the matrix. Here, each row corresponds to a potential pseudo-summary and its comparison with all other reviews. For each such pseudo-summary $r_i$, we select the top-k (first k with the highest scores) reviews $D_i$ and compute the mean scores $s_i$ across the selected reviews $D_i$.

| Dataset | Oposum+ | Amazon | Flipkart |
|---|---|---|---|
| No. of domains | 6 | 4 | 3 |
| No. of aspects | 18 | - | 38 |
| No. of test set | 60 | 32 | 147 |
| No. of reviews per product | 10 | 8 | 10 |
| No. of summaries per product | 3 | 3 | 1 |
| No. of general summaries | 180 | 96 | - |
| No. of aspect summaries | 540 | - | 676 |

**Table 4: Data statistics** for Oposum+, Amazon, and Flipkart test sets. Extractive summaries are underlined.

A synthetic pair $(D_i, r_i)$ is considered only if the score $s_i \geq$ threshold $\tau_g$. For our experiments, we set the value of $\lambda_1, \lambda_2, \tau_g$ and $k$ to be $0.5, 0.5, 0.38$ and $8$ respectively.

## 2.2 Aspect-specific Opinion Summarization

Synthetic dataset creation for aspect-specific opinion summarization can be divided into two phases: *aspect mapping* and *synthetic triplets creation*.

**Aspect Mapping** Prior methods relied on pre-defined mappings of seed words to aspects in order to initially filter sentences and assign them to corresponding aspects. However, these pre-defined mappings (Table 3) are seldom available. To address this limitation, we propose the automatic generation of such aspect mappings. Our approach begins with the utilization of an InstructABSA module (Scaria et al., 2023) to extract aspects. Subsequently, we employ sentence transformers and the fast clustering algorithm[1] (Reimers and Gurevych, 2019) to encode and cluster these aspects. We apply two distinct thresholds 0.7 and 0.9, to create coarse-grained and fine-grained clusters. The coarse-grained aspects function similarly to seed words, serving as filters for sentence selection, whereas the fine-grained aspects correspond to the aspects to which sentences will be mapped. Say, for a product, we get coarse-grained clusters $\{c_1, c_2, ...\}$ and fine-grained clusters $\{f_1, f_2, ...\}$. We map the coarse-grained cluster $c_i$ to fine-grained cluster $f_j$ using the condition $c_i \cap f_j \neq \phi$ to create the automatic aspect mapping (Table 3). **Appendix D**.

**Synthetic Triplets Creation** In the synthetic triplet creation phase, we construct triplets of the form {reviews, aspects, pseudo-summary} to facilitate supervised training of models that take reviews and aspects as input and the pseudo-summary as the output. In contrast to previous approaches, our

[1] https://bit.ly/fast-clustering

synthetic datasets also enable the use of multiple aspects for generating a summary. Let a review $r_i \in R$ ($R$ is the set of all reviews for a product) consist of sentences $\{x_{i1}, x_{i2}, ...\}$ and the sentence-level aspects are $\{a_{i1}, a_{i2}, ...\}$ where any $a_{ij}$ is a list of aspects. For each review $r_i$, we remove sentences $x_{ih}$ where $a_{ih} = \phi$ and concatenate sentences $x_{ij}$ and $x_{il}$ on the condition that $a_{ij} \cap a_{il} \neq \phi$. A review $r_i$, instead of sentences, can now be represented as consisting of portions (concatenation of sentences) $\{x'_{i1}, x'_{i2}, ...\}$ with corresponding aspect-lists $\{a'_{i1}, a'_{i2}, ...\}$. We replace each aspect with its coarse-grained aspect using the aspect mappings. Next, we create synthetic triplets. For a portion, say, $x'_{im}$ belonging to $r_i$, we concatenate all portions $x'_{tk} \in R_{-i}$ (where $R_{-i} = R - \{r_i\}$) to form $c_{im}$ such that $a'_{im} \cap a'_{tk} \neq \phi$. We also keep track of the reviews $R_{im}$ from $R_{-i}$ that contribute to the formation of $c_{im}$. Now, for each pair $(c_{im}, x'_{im})$, we compute a cosine similarity $ss_{im}$ and rouge score $rs_{im}$, and add them to form a unified score $us_{im}$ as $us_{im} = \lambda_1 * ss_{im} + \lambda_2 * rs_{im}$. We consider only those pairs where the unified score $us_{im} \geq$ threshold $\tau_a$. Finally, $(D_{im}, a'_{im}, x'_{im})$ forms a synthetic triplet, where $D_{im}$ is equal to $R_{im}$ or a combination of $R_{im}$ and some random reviews from $R_{-i} - R_{im}$ so that the final review set $D_{im}$ is limited to size $k$. For our experiments, we set the values of $\lambda_1, \lambda_2, \tau_a, k$ to $0.5, 0.5, 0.4$, and $10$ respectively. **Appendix E**.

## 3 Experiments

### 3.1 Datasets

We conducted experiments using three e-commerce datasets: Oposum+ (Amplayo et al., 2021), Amazon (He and McAuley, 2016; Bražinskas et al., 2020), and Flipkart (Siledar et al., 2023). The Oposum+ evaluation set contains extractive general summaries and aspect-specific abstractive summaries, Amazon contains general abstractive summaries, and Flipkart contains only aspect-specific abstractive summaries. Statistics are in Table 4. Using our SDC strategy, we generated 945K instances from Amazon for general opinion summarization whereas 350K instances from Oposum+ for aspect-specific opinion summarization. We test our models on Oposum+ and Amazon for general whereas Oposum+ and Flipkart for aspect-specific opinion summarization. **Appendix G**.

| | Model | asp? | Oposum+ | | | Amazon | | |
|---|---|---|---|---|---|---|---|---|
| | | | R1↑ | R2↑ | RL↑ | R1↑ | R2↑ | RL↑ |
| *Extractive* | Clustroid | ✗ | 33.44 | 11.00 | 20.54 | 29.27 | 4.41 | 17.78 |
| | LexRank | ✗ | 35.42 | 10.22 | 20.92 | 29.46 | 5.53 | 17.74 |
| | QT | ✗ | 37.72 | 14.65 | 21.69 | 34.04 | 7.03 | 18.08 |
| | Acesum$_{ext}$ | ✓ | 38.48 | 15.17 | 22.82 | x | x | x |
| | SW-LOO$_{ext}$ | ✓ | 40.45 | 19.13 | 23.20 | x | x | x |
| | NLI-LOO$_{ext}$ | ✓ | 39.79 | 18.33 | 23.49 | x | x | x |
| *Abstractive* | MeanSum | ✗ | 26.25 | 4.62 | 16.49 | 29.20 | 4.70 | 18.15 |
| | CopyCat | ✗ | 27.98 | 5.79 | 17.07 | 31.97 | 5.81 | 20.16 |
| | Acesum | ✓ | 32.98 | 10.72 | 20.27 | x | x | x |
| | SW-LOO | ✓ | _36.19_ | **12.17** | _21.11_ | x | x | x |
| | NLI-LOO | ✓ | 31.22 | 9.93 | 19.08 | x | x | x |
| | PlanSum | ✗ | 30.26 | 5.29 | 17.48 | 32.87 | 6.12 | 19.05 |
| | ConsistSum | ✗ | x | x | x | 33.32 | 5.94 | _21.41_ |
| | MultimodalSum | ✗ | 33.08 | 7.46 | 19.75 | 34.19 | 7.05 | 20.81 |
| | TransSum | ✗ | x | x | x | 34.23 | _7.24_ | 20.49 |
| | COOP | ✗ | x | x | x | **36.57** | 7.23 | 21.24 |
| | **Our Model** | ✗ | **36.57**[*] | 8.79 | **21.35** | _35.46_[*] | **7.30** | **21.50** |

**Table 5:** Evaluation for **general summaries** on Oposum+ and Amazon test sets. *asp?* indicates systems that use human-specified aspects. **Bold** and underline indicate best and second-best scores using abstractive systems.* indicates pvalue < 0.05 on paired t-test against MultimodalSum. Our model outperforms existing models on the task of general opinion summarization.

## 3.2 Baselines

**General Opinion Summarization.** Extractive approaches: *Clustroid* (Bražinskas et al., 2020), *LexRank* (Erkan and Radev, 2004), *QT* (Angelidis et al., 2021), *AceSum*$_{ext}$ (Amplayo et al., 2021), *SW-LOO*$_{ext}$ and *NLI-LOO*$_{ext}$ (Shen et al., 2023). For abstractive approaches we use *MeanSum* (Chu and Liu, 2019), *CopyCat* (Bražinskas et al., 2020), *Acesum, SW-LOO, NLI-LOO, Plansum* (Amplayo and Lapata, 2020), *ConsistSum* (Ke et al., 2022), *MultimodalSum* (Im et al., 2021), *TransSum* (Wang and Wan, 2021), and *COOP* (Iso et al., 2021).

**Aspect-specific Opinion Summarization.** We compare extractive approaches: *LexRank, QT, AceSum*$_{ext}$*, SW-LOO*$_{ext}$*,* and *NLI-LOO*$_{ext}$*.* For abstractive approaches we use *MeanSum, CopyCat, Acesum, SW-LOO, NLI-LOO,* and *ASBOS* (Siledar et al., 2023). **Appendix H.**

## 3.3 Implementation Details

We used `bart-large` (Lewis et al., 2019) as our Transformer model to fine-tune on our synthetic datasets. We used a learning rate of $2e-6$, batch size of 8 for 5 epochs. We performed manual hyperparameter tuning on dev sets to select the best model to report scores on the test set. During inference, we set the beam size to 2 and no repeat ngram to 3. For aspect-extraction, we use the

| | Model | asp? | Oposum+ | | | Flipkart | | |
|---|---|---|---|---|---|---|---|---|
| | | | R1↑ | R2↑ | RL↑ | R1↑ | R2↑ | RL↑ |
| *Extractive* | LexRank | ✗ | 22.51 | 3.35 | 17.27 | 10.41 | 0.93 | 8.72 |
| | QT | ✗ | 23.99 | 4.36 | 16.61 | 14.11 | 1.71 | 9.56 |
| | Acesum$_{ext}$ | ✓ | 26.16 | 5.75 | 18.55 | x | x | x |
| | SW-LOO$_{ext}$ | ✓ | 28.14 | 6.10 | 19.51 | x | x | x |
| | NLI-LOO$_{ext}$ | ✓ | 26.78 | 6.48 | 18.07 | x | x | x |
| *Abstractive* | MeanSum | ✗ | 24.63 | 3.47 | 17.53 | 10.64 | 1.33 | 9.78 |
| | CopyCat | ✗ | 26.17 | 4.30 | 18.20 | 13.48 | 1.92 | 10.35 |
| | Acesum | ✓ | 29.53 | 6.79 | _21.06_ | x | x | x |
| | SW-LOO | ✓ | _30.00_ | **6.92** | 20.76 | x | x | x |
| | NLI-LOO | ✓ | 28.90 | 6.60 | 20.11 | x | x | x |
| | ASBOS | ✓ | 23.45 | 4.37 | 16.85 | _14.62_ | _2.23_ | _11.56_ |
| | **Our Model** | ✗ | **30.95** | **6.92** | **21.73** | **20.15** | **2.86** | **15.99** |

**Table 6:** Evaluation for **aspect summaries** on Oposum+ and Flipkart test sets. *asp?* indicates systems that use human-specified aspects. **Bold** and underline indicate best and second-best scores using abstractive systems. Our model without relying on any human-specified aspects or seed words learns the task of aspect-specific opinion summarization and outperforms alternatives.

`kevinscaria/ate_tk-instruct-base-def-pos-neg-neut-combined` (Scaria et al., 2023) model. For encoding reviews we use the `sentence-transformers/all-MiniLM-L12-v2` (Reimers and Gurevych, 2019). **Appendix J.**

## 4 Results and Analysis

**General Opinion Summarization.** We use the ROUGE-{1,2,L} F1 score (Lin, 2004) (R1, R2 & RL) to assess the generated summary quality. Table 5 reports results on general opinion summarization. On both Oposum+ and Amazon, our model outperforms MultimodalSum. Overall, our model achieves the best R1 and RL on Oposum+ and the best R2 and RL whereas second-best R1 on the Amazon test set. Our synthetic dataset creation approach plays an essential role in our models performing better. By meticulously selecting {input reviews, pseudo-summary} pairs that are lexically and semantically most similar to one another, we enable our models to effectively learn the task of opinion summarization. Additionally, this also ensures that the model generates summaries that are most faithful to the input reviews.

**Aspect-specific Opinion Summarization.** Table 6 presents results on aspect-specific opinion summarization. Without using any human-specified aspects and seed words, our model is able to achieve the best scores on both datasets across all metrics. As a result of automatic aspect mappings, our syn-

| Model | SummaC ↑ | CTC ↑ | FactCC ↑ | FactGraph ↑ |
|---|---|---|---|---|
| PlanSum | 0.33 | 0.81 | 0.16 | 0.21 |
| Multimodalsum | 0.38 | **0.85** | 0.46 | 0.53 |
| **Our Model** | **0.40** | **0.85** | **0.68** | **0.66** |

**Table 7: Faithfulness Evaluation.** Our model outperforms alternatives on three faithfulness measuring metrics: SummaC, FactCC, and FactGraph.

thetic datasets contain training points with a significantly broader range of aspects compared to the pre-defined ones. This diversity in aspects ensures that the models trained on our synthetic datasets are better equipped to effectively learn the nuances of aspect-specific opinion summarization. Finally, as Flipkart contains only a test set, we use models trained using the Oposum+ dataset for testing.

**Analysis.** For general opinion summarization, our model-generated summaries usually contain information that is verifiable from the input reviews, e.g., the color of the teapot being *dark turquoise* (Table 1). This we attribute to our SDC strategy for general synthetic datasets. For aspect-specific opinion summarization, our model is able to generate aspect-specific summaries without ever being explicitly trained on aspects, e.g., *ports* (Table 2). This we intuit is possible as our model is trained on a wide range of automatically identified aspects instead of a few pre-defined ones. Finally, we do observe that aspect-specific summaries have a tendency to generate information irrelevant to the aspect under consideration.

**Faithfulness Evaluation.** We use four automatic faithfulness evaluation metrics: SummaC (Laban et al., 2021), CTC (Deng et al., 2021), ,FactCC (Kryscinski et al., 2019), and FactGraph (Ribeiro et al., 2022) on the Amazon test set for general opinion summarization. Table 7 shows that our model scores higher than the other models on all the metrics indicating that the summaries generated from our models are much more faithful to the input reviews compared to others. **Appendix K.**

**Ablation Study.** We conduct ablation studies to analyze the impact of using different metrics and filtering strategies. Specifically, we compare results using either cosine similarity or R1 alone, finding that the scores decrease when only one metric is used. Furthermore, we assess the effectiveness of our pseudo-summary selection approach by

| Our Model | OPOSUM+ | | Amazon | Flipkart |
|---|---|---|---|---|
| | Aspect | General | General | Aspect |
| *w/ Both metrics* | **21.73** | **21.35** | **21.50** | **15.99** |
| *w/ Cosine Similarity* | 20.25 | 20.07 | 19.48 | 15.60 |
| *w/ R1* | 20.50 | 20.23 | 19.90 | 15.63 |
| *w/ Training Random* | 17.85 | 16.06 | 16.18 | 7.04 |

**Table 8: Ablation Study.** *Cosine Similarity* or *R1* indicates the use of only that metric. *Training Random* means randomly selecting reviews and a pseudo-summary. We report the RL scores on the test sets.

| Amazon | Faithfulness ↑ | Coherent ↑ | Concise ↑ | Fluency ↑ |
|---|---|---|---|---|
| PlanSum | -0.92 | -0.91 | -0.84 | -0.72 |
| Multimodalsum | 0.41 | **0.45** | 0.40 | 0.31 |
| **Our Model** | **0.51** | **0.45** | **0.41** | **0.41** |

**Table 9: Best-Worst Scaling.** Our model receives better scores on three criteria. Best values are in **bold**.

training models with random reviews and pseudo-summary and evaluating its performance on the test set. Table 8 demonstrates our choice of metrics in achieving improved performance.

**Human Evaluation.** We used Best-Worst Scaling (Louviere et al., 2015) to assess the quality of general opinion summaries. Three participants evaluated model summaries on *faithfulness, coherence, conciseness,* and *fluency*. They compared the model summaries to human-written summaries from the Amazon test set and selected the best and worst summaries accordingly. Table 9 shows that our model outperformed alternatives across all criteria. **Appendix L.**

## 5 Conclusion and Future Work

We proposed synthetic dataset creation approaches for general and aspect-specific opinion summarization. For general opinion summarization, our SDC approach enables models to generate summaries that are more faithful to the input than existing approaches, as indicated through automatic and human evaluations. For aspect-specific opinion summarization, our SDC approach enabled models to outperform existing models without relying on any human-specified aspects or seed words, making our approach generalizable to other opinion summarization domains where such human-specified aspects or seed words are not defined.

We plan to further extend our work to domains other than e-commerce for opinion summarization.

## Limitations

We showed results only on e-commerce datasets. However, the synthetic dataset creation approaches proposed here are not domain specific and are generalizable to domains other than e-commerce. The other challenge specific to opinion summarization is to summarize a large number of reviews. It is infeasible to input all the reviews to train a model for summarization, limited by modern hardware capacity. As a result, existing approaches either resort to a hybrid approach of extractive filtering followed by abstractive summarization or use a smaller set of input reviews for summarization. We resort to the second approach to make comparisons with existing approaches. Finally, we did not make comparisons with recent large language models for opinion summarization as our focus was on pushing for improvement in the synthetic dataset creation approaches to train smaller models and also to perform a fair comparison to existing approaches.

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

# A  Related Work

Earlier research in the field of opinion summarization has focused on extractive approaches (Hu and Liu, 2006; Kim et al., 2011; Angelidis and Lapata, 2018) as well as abstractive approaches (Ganesan et al., 2010; Carenini et al., 2013; Di Fabbrizio et al., 2014). The absence of large amounts of annotated opinion summarization datasets, recent works consider the unsupervised setting for opinion summarization where only review corpus is available without the corresponding summary (Chu and Liu, 2019; Bražinskas et al., 2020).

## A.1  Aspect-specific Opinion Summarization

Angelidis et al. (2021) proposed the first approach to generate both aspect-specific and general summaries. They utilize a Vector Quantized Variational Autoencoder (Van Den Oord et al., 2017) for clustering review sentences followed by a popularity-driven extraction algorithm to summarize. (Basu Roy Chowdhury et al., 2022) utilizes dictionary learning (Dumitrescu and Irofti, 2018) to acquire representations of texts based on latent semantic units. Amplayo et al. (2021) proposed the first abstractive approach for generating aspect-specific and general summaries. They generate synthetic datasets by identifying aspect-bearing elements (words, phrases, sentences) using a multiple instance learning (MIL) (Keeler and Rumelhart, 1991) model trained on silver-labeled data obtained through seed words. Shen et al. (2023) proposes two simple solutions for generating synthetic datasets that do not rely on complex MIL modules. The SW-LOO simply matches the aspect seed words to construct synthetic datasets, whereas NLI-LOO uses an off-the-shelf NLI model to do so using only aspects and no seed words. Mukherjee et al. (2020) takes an unsupervised approach to extract aspects and manually create a mapping between fine-grained and coarse-grained aspects using Integer Linear Programming (ILP) based extractive subset of opinions. Our work is closest to (Shen et al., 2023) in that we also build synthetic datasets without the use of complex MIL models. However, we are different in generating synthetic datasets as our approach does not rely on any human-specified aspects or seed words. We use off-the-shelf aspect extraction and clustering techniques to build an automatic mapping of aspects. Moreover, our approach uses two metrics: cosine similarity and rouge scores to form synthetic datasets which achieves better performance. Our approach of aspect mapping is similar to Mukherjee et al. (2020) where we create a mapping between fine-grained and coarse-grained aspects. However, our approach is fully automatic and does not need any human intervention to perform the mapping.

## A.2  General Opinion Summarization

Chu and Liu (2019); Bražinskas et al. (2020) use autoencoders (Kingma and Welling, 2013) and its variants to learn a review decoder through reconstruction which is then used to generate summaries using the averaged representations of input reviews. Another approach is to curate synthetic datasets using one of the reviews as a pseudo-summary and pair it with input reviews using different strategies. Bražinskas et al. (2020) uses random sampling, Amplayo and Lapata (2020) generates noisy version of the pseudo-summary, Elsahar et al. (2021) ranks reviews using similarity and relevance, and Amplayo and Lapata (2020) uses content plans to generate synthetic datasets. Im et al. (2021) randomly selects a review as a pseudo-summary and proposes a pipeline to generate summaries using multimodal input sch text, image, and meta-data. Ke et al. (2022) captures the consistency of aspects and sentiment between reviews and summary, whereas Wang and Wan (2021) learns aspect and sentiment embeddings to generate relevant pairs. Iso et al. (2021) searches for convex combinations of latent vectors to generate summaries. Our approach uses cosine similarity and rouge scores between reviews to filter highly relevant synthetic pairs that enable models to generate summaries more faithful to input reviews compared to alternative approaches.

```
"Three of us stayed at the Hotel Navona and shared a room.",
"It had two twins pushed together and a separate cot-like bed.",
"We all found the room to be more than large enough and very comfortable.",
"I slept on the cot-like bed and slept very well.",
"Shower worked well.",
"Our room was on a courtyard with a large, heavy oak door.",
"You cannot beat the atmosphere and location.",
"The staff is very helpful and friendly.",
"Location, Location, Location!"
```

**Figure 1:** Sample reviews for a hotel domain.

## B Generalizability

In this work, we focus on only e-commerce datasets and show results on them. However, our approaches are generalizable and could be used for any other domain. Because our approaches do not rely on human-specified aspects or seed words, it can be extended to any domain. Figure 1 shows an example of reviews for a hotel. Using our approach, we will be able to identify aspects such as *room, atmosphere, location, staff, ...* and create synthetic pairs corresponding to it from the training data.

## C Opinion Summarization Model

For general opinion summarization, only concatenated reviews are provided as input in the following format:

```
 review₁  review₂  ... 
```

For aspect-specific opinion summarization, we transform the aspects and reviews into the following format:

```
 aspect₁, aspect₂, ... 
 review₁  review₂  ... 
```

where `` and `` are special tokens for BART and aspects are comma-separated.

## D Aspect Mapping

The primary idea behind this approach is to generate something similar to human-specified aspects and seed words. As an illustration, let's consider the laptop bag product from the Oposum+ dataset as shown in Table 3, where the aspect *looks* is associated with seed words such as *looks, color, stylish, looked,* and *pretty*. Sentences from reviews are filtered for the *looks* aspect if any of these seed words appear in them. We establish similar mappings by employing aspect extraction and clustering techniques as shown in Table 3. Our approach involves

forming clusters at two levels: fine-grained (using a low threshold) and coarse-grained (using a high threshold). For instance, fine-grained aspects like *camera, selfie camera, camera quality, back camera* should all be mapped to the coarse-grained aspect *camera* by associating them with the cluster that contains *camera*. The aspect mapping in Table 3 demonstrates the final mapping for a sample product. Now, we can utilize this automatically generated mapping to filter review sentences and create synthetic datasets.

## E SDC for Aspect-specific Opinion Summarization

To begin, we identify the sentence-level aspects for each product using aspect mappings. We then eliminate sentences that do not contain any aspects. Within a review, if there are sentences that share common aspects, we merge them together and refer to them as portions. The objective is to determine if a specific portion is of sufficient quality to be considered a pseudo-summary. We combine all remaining portions that share the same aspects with the current portion under consideration and calculate cosine similarity and rouge scores between them. This process is repeated for all portions across all reviews associated with a particular product.

## F Why our models perform better?

**General Opinion Summarization** In opinion summarization, synthetic dataset curation is important due to the absence of any large-scale supervised datasets (reviews-summary pairs). Such synthetic datasets enable supervised training for models. In Appendix A, we discussed how existing approaches create synthetic datasets followed by the generation of opinion summaries. We, however, hypothesize that a set of reviews and a pseudo-summary can act as a potential pair only if the pseudo-summary is lexically (R1) and semantically (cosine similarity) similar to the review set. We use the matrix construction approach to find such pairs which ensures that only highly relevant pairs are selected in our synthetic dataset. The simple intuition behind this is that during training we want to show the model a pseudo-summary that is most relevant to the input reviews. This will enable the model to learn the task of opinion summarization much more accurately. This is evidenced by results

in Table 5 as well as faithfulness scores in Tables 7 and 9 in the paper.

**Aspect-specific Opinion Summarization** Existing approaches are dependent on human-specified (pre-specified) aspects and seed words for creating synthetic datasets during training and filtering sentences from reviews before generating summaries during inferencing. Our approach, however, creates aspect mappings automatically which enables the creation of synthetic datasets for a wide range of aspects. Here as well we use the combination of lexical and semantic scores for selecting a pseudo-summary (set of sentences about an aspect) and reviews pair. This ensures that the most relevant review-pseudo summary pair is shown to the model during training enabling better learning for the model to generate summaries. Our synthetic datasets are not restricted by the human-specified aspects and seed words which makes the model more robust in generating summaries for any relevant aspect of a specific product. This we intuit is causing improved scores of our models over existing models even though existing models have this extra aid of human-specified aspects and seed words for training and inference.

## G  Dataset Details

**Oposum+**  Oposum+ contains product reviews from six different domains: *laptop bags, bluetooth headsets, boots, keyboards, televisions* from the *Amazon Product Dataset* (He and McAuley, 2016). Evaluation set contains four summaries per product: three aspect-specific abstractive summaries and one general extractive summary. Each product has 10 reviews in the evaluation set. The training set contains ∼ 4.13M reviews over 95K products.

**Amazon**  Amazon dataset contains product reviews from four domains: *electronics, home and kitchen, personal care,* and *clothing, shoes and jewelry*. The evaluation set contains three general abstractive summaries per product. Each product has 8 reviews in the evaluation set. The training set contains ∼ 1M reviews over 90K products.

**Flipkart**  Flipkart dataset contains product reviews from three domains: *laptops, mobiles*, and *tablets*. The test set contains around 147 products with one summary per product. Each

summary consists of multiple aspect-specific summaries. There are around 676 aspect-specific summaries in total. The original test set contains around 1000 reviews per product on average. We downsample this to 10 reviews per product to compare different models. We first remove all the reviews with less than 20 and more than 100 words. For filtering out 10 reviews we use a simple approach of first checking if the reviews contain the aspects for which summaries need to be created. After the filtering step, we randomly selected 10 reviews to form input for our test set.

## H  Baseline Details

**General Opinion Summarization** *Clustroid* is a simple extractive model that selects the review with the highest RL score with respect to other reviews. For abstractive approaches, we use *AceSum* which generates synthetic datasets with the help of a multiple-instance learning model. *SW-LOO* and *NLI-LOO* use the leave-one-out strategy to construct synthetic datasets. *PlanSum* used content plans to generate synthetic datasets by extracting a representative review as the pseudo summary. *ConsistSum* uses aspect and sentiment distribution to generate review-summary pairs. *MultimodalSum* proposes a pipeline to generate summaries using multimodal data such as text, images, and meta-data. *TransSum* uses aspect and sentiment embeddings to construct synthetic datasets to train a supervised opinion summarization model. *COOP* searches for convex combinations of latent vectors to generate summaries.

**Aspect-specific Opinion Summarization** *LexRank* selects the most salient sentences using BERT (Devlin et al., 2019) encodings to represent sentences. *QT* is a neural clustering method that represents opinions in quantized space. *AceSum$_{ext}$* is the extractive version that uses sentences ranked by their controller induction model as input to *LexRank*. Similarly, *SW-LOO$_{ext}$* and *NLI-LOO$_{ext}$* also input aspect-related sentences to *LexRank*. Abstractive approaches such as *MeanSum* generate summaries by reconstructing the mean of the encoded reviews. *CopyCat* is

a hierarchical variational autoencoder that learns a latent code of the summary. *ASBOS* uses a hybrid approach of filtering review sentences on the basis of aspects and then summarizing them.

## I  Training Details

In Table 6, results for the Flipkart test set are obtained from models trained using the Oposum+ training data due to the unavailability of any training data. In Table 5 and 6, results that couldn't be produced due to the unavailability of public code or certain issues have been marked with x.

## J  Implementation Details

We use BART implementation from HuggingFace (Wolf et al., 2019). We use Adam (Kingma and Ba, 2015) optimizer with eps of $1e - 4$ and linear weight decay to optimize our models. We use $[1e - 6, 2e - 6, 1e - 5, 2e - 5]$ and batch size $[8, 16]$ as our hyperparameters. We observe that a learning rate of $2e - 6$ and batch size of $8$ performs the best. All experiments were performed using NVIDIA A100-SXM4- 80GB GPUs. Refer Section 3.3.

## K  Faithfulness Metrics Details

We measure the faithfulness of summaries using four faithfulness measuring metrics: SummaC, CTC, FactCC, and FactGraph. We chose these metrics for evaluation as they correlate better with human judgments (Chaudhury et al., 2022).

**SummaC**: SummaC (Summary Consistency) aims to tackle the granularity in NLI models. Specifically, SummaC focuses on identifying inconsistencies in summarization, taking into account the diverse levels of granularity that can exist between sentences and documents A higher SummaC score indicates higher faithfulness.

**CTC**: CTC (Compression Transduction Creation) presents a framework that considers various natural language generation (NLG) tasks, including compression (such as summarization), transduction (like text rewriting), and creation (such as dialog generation). The CTC metric evaluates information alignment, with a specific emphasis on gauging consistency and rel-

evance. A higher CTC score indicates higher faithfulness.

**FactCC**: FactCC is a BERT-based classification model, with the objective of ascertaining the consistency or inconsistency between a provided text or summary and its corresponding source article. A higher FactCC score indicates higher faithfulness.

**FactGraph**: FactGraph uses both the text and their structured meaning representations computed using a graph encoder with structure-aware adapters to enhance the factuality of the summaries with respect to the source document. A higher FactGraph score indicates higher faithfulness.

## L  Best Worst Scaling

We evaluated our model-generated general opinion summaries on four different criteria: *faithfulness* (how consistent are the opinions compared to reference summaries?), *coherence* (is the summary well organized and easy to read?), *conciseness* (is the summary concise yet informative?), and *fluency* (is the summary fluent and grammatical?) (Amplayo et al., 2021). Each participant's choice of the best model earned a score of +1, while the worst model received a score of -1. The remaining models were assigned a score of 0. To calculate the final scores, we averaged the scores from all participants' responses. Table 9 indicates that our model gives the best scores on all the criteria. Specifically our model scores around 0.1 point higher than the second-best model indicating that our model-generated general opinion summaries are more faithful than alternatives.

## M  Example Summaries

Table 10 shows sample model-generated summaries for both general opinion summarization and aspect-specific opinion summarization. For general opinion summarization, we show three human-written summaries followed by model-genereated summaries compared to baselines for a product from the Amazon test set. For aspect-specific opinion summarization, we show human-written summaries followed by model-generated summaries compared to baseline on different aspects for a product from the Oposum+ test set.

**General Opinion Summarization**

**Human written summaries**

These look very nice on. The mesh in them run a little tight though so you may want to go up in size. They are roomy in the pockets and can hold onto a lot of items. They are a great price for the quality and I would highly recommend them to anyone looking for a new pair of swim trunks.

These swim trunks look great and the waist fits as expected for the size. They are also a good deal for the price. However, the inner mesh liner is much smaller than the size of the outer trunks. You will probably have to cut out the mesh liner to wear them comfortably.

These are nice fitting shorts but the mesh liner felt tight and I cut it out. You might want to size up if you are unsure. The deep pockets come in handy. These shorts are stylish and they look good. They are more comfortable if the inner mesh liner is removed.

**PlanSum**

i bought this for my husband and she loves it. it is a very good product and the fit is good. the only reason i didn't give it 5 stars is because it does not fit in the middle of the back of my shirt. i would recommend this to anyone who is looking for a good quality product.

**MultimodalSum**

I bought these for my husband to wear to the beach. They fit well and look good. The only thing I don't like is that the mesh liner is a little tight. He has to pull it up every time he goes in and out of the water.

**Our Model**

These shorts are great. They fit well and look great. The only thing I would change is the liner. It is a little too small and I had to cut it out. Other than that, I would recommend these shorts.

**Aspect-specific Opinion Summarization**

**Human written summaries**

_Ports_ - in some cases the output headphones and the hdmi ports have stopped working , however it has enough inputs and for the price it is a pretty solid option .

_Picture_ - the quality is good and the resolution is decent . sharpness is generally good , although blacks are somewhat inconsistent in the image .

_Sound_ - the sound quality is acceptable for the price of the equipment , the sound can be described above average . may not need a sound bar .

**AceSum**

_Ports_ - the hdmi ports stopped working and the tv would go black for about three to five seconds every few minutes. it has all of the inputs for us, and more.

_Picture_ - the picture quality was better than my 5 year old lcd samsung. the picture is decent, but not as good as the htpc.

_Sound_ - i bought the tv at fry's after salesperson assured me headphones would work with this. picture is acceptable and sound is above average for the price.

**Our Model**

_Ports_ - i have had this tv for a few months now and i have had no issues with it, it has all of the hdmi ports i need and the picture quality is great.

_Picture_ - the picture is great, the tv is easy to set up, and it has all of the inputs i need, but it is not a smart tv.

_Sound_ - the sound is decent, but i have a surround sound system so i don't use the tv's speakers much, but it is a great tv for the price.

**Table 10: General Opinion Summarization:** Three human-written summaries followed by summaries from different models for a product from the Amazon test set for general opinion summarization. **Aspect-specific Opinion Summarization:** Human written summaries followed by summaries for a product on aspects: _Ports,_ _Picture,_ and _Sound_ from the Oposum+ test set for aspect-specific opinion summarization.