# OpenReview forum: "Synthesize, if you do not have: Effective Synthetic Dataset Creation Strategies for Self-Supervised Opinion Summarization in E-commerce"
_EMNLP/2023/Conference — EMNLP 2023 Findings_

### Official Review · Reviewer_GYEt · 2023-07-23

**Soundness:** 3

**Excitement:**

3: Ambivalent: It has merits (e.g., it reports state-of-the-art results, the idea is nice), but there are key weaknesses (e.g., it describes incremental work), and it can significantly benefit from another round of revision. However, I won't object to accepting it if my co-reviewers champion it.

**Missing References:**

- Some of the baseline systems in 3.2 don't have references in place.

**Paper Topic And Main Contributions:**

This paper proposes methods for creating synthetic training sets for generic and aspect-based review summarization.
- For aspect-based summarization, an aspect mapping is first prepared using a two-level clustering method (on SentenceBERT representations of aspects extracted by the off-the-shelf InstructABSA). The fine-grained "aspects" in review sentences can be mapped to more general coarse grained ones. Then "portions" of reviews are prepared for each coarse-grained aspect, and partial reviews are grouped together with related full reviews to form triplets of reviews, aspects and a pseudo-summary.
- For generic summarization, reviews that are most similar to other reviews on ROUGE and cosine-similarity of sentenceBERT are set as pseudo-summaries for the others.
- A BART-based model is trained on each of these datasets, and compared to several baselines on Amazon and Flipkart product reviews. The model is generally the best overall in ROUGE and in three faithfulness metrics. It is also best overall in a manual evaluation against two other relatively strong baselines on faithfulness, coherence, conciseness and fluency.

**Reasons To Accept:**

- A new way to create synthetic data for aspect-based summarization.
- With a relatively simple model, the training data enables high scores against previous methods.

**Reasons To Reject:**

There are several points that were hard for me to follow:
- The reference to "human-specified aspects" and "seed-words" already in the abstract was not clear. Usually aspect-based summarization has a set of pre-specified aspects for a product-type. Case-specific aspects are known as "open aspects", and this is not what this paper seems to describe (e.g. for evaluation on Oposum+). It should be made clearer that by having a train set that is not aspect-specific, a model may be able to be trained robustly for any aspect.
- The motivation that there are faithfulness issues in other synthetic datasets is not clear to me. There is nothing proving this or even giving an intuition for why it might be true. The fact that the paper's baseline gets higher scores on faithfulness doesn't imply that there is a problem in other synthetic datasets.
- There's no explanation on the "fast clustering algorithm" and the thresholds for the granularity of aspects. It should at least be added in appendix E, since this is supposedly part of an important contribution of the paper.
- The whole "Synthetic Triplets Creation" paragraph is very difficult to follow due to the way it is phrased and formalized with notations. E.g.: (1) Line 120: what is "combined"; (2) the term "portion" is not intuitive without deeply apprehending what is going on; (3) why are r_i1 and r'_t5 used, and not generic indices?; (4) Line 130: what is "concatenate"?; (5) Line 131: what is c in c_i1 notation?; (6) Line 134: Between what two instances is cossim and rouge computed?; (7) how are the hyperparameters chosen?; (8) Line 145: why are there two aspects? the standard aspect-based summary only considers one? If this is an extension somehow, it should be explained.
- Section 2.2 also has some unclear points: (1) Line 168: I have to guess what top-k is; (2) Line 169: What are the "selected" reviews?; (3) Line 173: What is D_i?; (4) Again, how are hyperparams chosen? (5) What is the final dataset?; (6) Generally, the approach for synthesizing generic pseudo-summaries (using a combination of two classic similarity metrics) is a well-known method and was used in some of the papers referenced and others. If there is something new about the approach it should be explicitly mentioned.
- Overall, I don't really understand what the two final datasets contain per instance.
- In Section 3.1, it's not clear how there are so many instances created when the source datasets are so small. Table 2 talks about a few hundred instances, but the synthetic data is of hundreds of thousands. The appendix says something about it, but even there, it's not clear what the train set of these datasets are.
- In section 3.3 or in Section 4, it doesn't say anything explicit about the baseline being trained on the synthetic datasets or how (e.g. when aspects are present).
- Line 233: it says "even though [the Oposum+ variant] is trained on Amazon dataset", but Oposum+ *is* based on Amazon data, unless I'm missing something, so it is as expected.
- Line 243: The automatic faithfulness metrics do not "prove" the claim that the summaries are more faithful, they only "indicate" it. These metrics are known to be very approximate. Also, is this on the aspect-based or generic summarization? it doesn't say.
- While most of the relevant related work is in the introduction, some of it is in the appendix, and would be beneficial in the main paper.
- I think the comparison to "random training data" in the ablation in Appendix D is important to further convince the reader that the training set is indeed worthwhile. It's hidden away in the appendix, and should be mentioned in the paper itself.
- Some presentation points confused me (see in separate box)

Generally, it's hard to tell what actually causes the improvement against the baselines. Is it the BART-based model? Is it the abundance of the training data? Is it the quality of the data? The comparison is not so fair since the baseline models are not trained on the same data. If the point is to show that the proposed synthetic data is better and also improves faithfulness, the convincing way to do it would be to train several models with different training data separately (and taking quantity of training instances into account), and comparing the results on that. The storyline of this current version of the paper would be convincing that way much more. Otherwise, the paper can be motivated differently and then corresponding proper experiments would be required.

Also, since aspects are prepared on-the-fly, it says that the proposed strategy enables creating synthetic aspect-based summarization data for other domains as well. But I'm not sure about that because reviews and, e.g., news, have very different behavior when pseudo-summaries are the main player. This assumption needs to be checked.

Sorry for the many comments, I'm trying to provide constructive points to improve the paper, at least in my eyes.


------

UPDATE: I have raised my scores from 2/2 to 3/3, hoping that the authors will indeed update the paper as they have suggested in the rebuttals.

**Reproducibility:**

2: Would be hard pressed to reproduce the results. The contribution depends on data that are simply not available outside the author's institution or consortium; not enough details are provided.

**Reviewer Confidence:**

3: Pretty sure, but there's a chance I missed something. Although I have a good feel for this area in general, I did not carefully check the paper's details, e.g., the math, experimental design, or novelty.

**Typos Grammar Style And Presentation Improvements:**

- "General" and "aspect-specific" summarization are usually termed "generic" and "aspect-based" summarization. This confused me at first, that maybe this paper was regarding different tasks that I was unaware of.
- Line 43: Text summarization does not "contain" reference summaries, rather a dataset does.
- Figure 1: The "filtered-reviews using scores" diagram is not clear, even after reading the whole paper, and especially not when the figure is first referenced.
- Second contribution bullet in intro: Explicitly say that this is for generic summarization.
- Line 118: "... *consist* of sentences and the *corresponding* sentence-level...
- Paragraph titles are usually not in title case, and end with a period.
- Line 161: "unified" is not so clear, just "combined"

---

> ### Author Rebuttal · Authors · 2023-08-29
>
> We thank the reviewer for the valuable suggestions and feedback. We have tried our best to resolve all the doubts and provide clarifications. We request the reviewer to kindly consider increasing the scores positively if they find the rebuttal satisfactory.
>
> ### A. Additional Results
>
> **1. General Opinion Summarization:** Due to certain issues with our code, our results in the paper are with beam size 1. We re-ran our updated code with different beam sizes and observed the best performance using beam size 5 (decided using the dev set). Currently, we achieve the best R2 and RL score and second-best R1 compared to baselines.
>
> | *Amazon* | R1 | R2 | RL |
> | ------------- | ---- | ---- | ---- |
> | Multimodalsum | 34.19 | 7.05 | 20.81 |
> | ConsistSum | 33.32 | 5.94 | _21.41_ |
> | TransSum | 34.23 | _7.24_ | 20.49 |
> | COOP [1] | **36.57** | 7.23 | 21.24 |
> | Ours | _35.46*_ | **7.30** | **21.50** |
>
> **Rebuttal Table 1:** **Bold** and _italics_ indicate best and second-best results. *indicates results that are significantly better than MultimodalSum with a pvalue<0.05 on paired t-test computed over the Amazon test set for general opinion summarization (couldn't be compared with others due to unavailability of code/results).
>
>
> **2. Aspect-specific Opinion Summarization:** Due to certain issues with our code, our results in the paper are with beam size 1. We re-ran our updated code with different beam sizes and observed the best performance using beam size 2 (decided using the dev set). Currently, we achieve the best R1, R2, and RL scores compared to baselines.
>
> | *Oposum+* | R1 | R2 | RL |
> | ------------- | ---- | ---- | ---- |
> | Acesum | 29.53 | 6.79 | _21.06_ |
> | SW-LOO | _30.00_ | **6.92** | 20.76 |
> | NLI-LOO | 28.90 | 6.60 | 20.11 |
> | Ours | **30.95*** | **6.92** | **21.73** |
>
> **Rebuttal Table 2:** **Bold** and _italics_ indicate best and second-best results. *indicates results that are significantly better than AceSum with a pvalue<0.05 on paired t-test computed over the Oposum+ test set for aspect-specific opinion summarization (couldn't be compared with others due to unavailability of code/results).
>
>
> | Acesum |  |
> | --------- | ---------------- |
> | Ports | the hdmi ports stopped working and the tv would go black for about three to five seconds every few minutes. it has all of the inputs for us, and more. |
> | Picture | the picture quality was better than my 5 year old lcd samsung. the picture is decent, but not as good as the htpc.|
> | Sound | i bought the tv at fry's after salesperson assured me headphones would work with this. picture is acceptable and sound is above average for the price. |
> | **Ours** |  |
> | Ports | i have had this tv for a few months now and i have had no issues with it, it has all of the hdmi ports i need and the picture quality is great. |
> | Picture | the picture is great, the tv is easy to set up, and it has all of the inputs i need, but it is not a smart tv. |
> | Sound | the sound is decent, but i have a surround sound system so i don't use the tv's speakers much, but it is a great tv for the price.|
>
> **Rebuttal Table 3:** Comparative analysis of our model-generated summaries with Acesum for aspect-specific opinion summarization on different aspects such as _ports, picture,_ and _sound_. Our model does not rely on any human-specified aspects and seed words for training, making our approach generalizable to other domains where such aspects and seed words are not defined.
>
> **3. Faithfulness Evaluation:**  We additionally show results of general opinion summarization on FactGraph [6] which is a metric that correlates the best with human judgment on faithfulness evaluation [2].
>
> |  | FactGraph |
> | ------------- | ---- |
> | Plansum | 0.16 |
> | Multimodalsum | 0.53 |
> | Ours | **0.66** |
>
> **Rebuttal Table 4:** **Bold** indicate the best result. We achieved the highest score on FactGraph for faithfulness evaluation for general opinion summarization.
>
> | Models | Summaries |
> | --------- | ---------------- |
> | Multimodalsum | This is a good receiver for the price. I have it **hooked up to a pair of Sony speakers** and it sounds great. The only problem I have with it is that the **remote control is a little difficult to use**. I would recommend this to a friend. |
> | Ours | This is a great stereo for the price. The only thing I don't like about it is the fact that you can't adjust the volume on the radio. I have to turn it up all the way to hear it. Other than that, it's a great product. |
> | Multimodalsum | This is a nice teapot, but the color is not as bright as the picture. It is more of a dark turquoise than a light blue. **I was hoping it would be more of an aqua blue, but it is more like a dark aqua**. It still looks nice, but I would have preferred the color to be more like the photo. |
> | Ours | This is a beautiful teapot, but the color is not the same as shown in the picture. It is more of a dark turquoise. The color is a little darker than the picture, but it is still a beautiful color. The only thing I don't like about it is that the handle is very hard to open.|
>
> **Rebuttal Table 5:** Example of faithfulness issue in baseline models compared to ours. There is no support for the information in **bold** parts in the corresponding input.
>
> ### B. Clarifications on the naming convention for General, Aspect-specific, human-specified aspects, and seed words
> Sorry for any confusion this might have caused, but we did use standard terminologies following existing literature ([3], [4]) for general and aspect-specific opinion summarization. Seed-words is also a standard term in aspect-specific opinion summarization ([3], [4]). Human-specified aspects are the aspects defined for a product category (eg. Bags) by human annotators (eg. looks, quality, size) [1]. We agree that this point should have been made clearer in the paper.
>
> ### C. Clarifications on our approach applicable to other domains
>
> By other domains, we specifically meant domains within opinion summarization such as hotels (example in Appendix B), movies, businesses, etc. which contain opinions in the form of reviews. We do agree that domains falling under text summarization such as “news” have different behaviors and the applicability of these approaches needs to be investigated separately. (Will update this clarification in the paper)
>
> ### D. Confusion regarding Datasets
>
> Amazon and Oposum+ although based on Amazon data [5] are named as such in existing literature. Hence separate mentions for Oposum+ and Amazon to keep things clear. The main difference between these two datasets is the domain they cover, mentioned in Appendix G. Specifically, Oposum+ covers _laptop bags, bluetooth headsets, boots, keyboards, televisions_ whereas the Amazon dataset [5] covers _electronics, home and kitchen, personal care, and clothing, shoes and jewelry_.
> Table 2. stats are for the Amazon, Oposum+, and Flipkart test sets. For training, we create synthetic datasets out of the review corpus, statistics for which have been mentioned in Appendix G. Specifically, we create 945k instances from ∼1M reviews over 90k products in the Amazon dataset for general opinion summarization. We create 345k instances from ∼4.13M reviews over 95k products in the Oposum+ dataset for aspect-specific opinion summarization.
>
> Each synthetic dataset instance for general opinion summarization contains the following:
>
> 1. Input reviews (8 for Amazon, 10 for Oposum+ and Flipkart)
>
> 2. Pseudo-summary (Selected through our synthetic dataset creation strategy)
>
> Each synthetic dataset instance for aspect-specific opinion summarization contains the following:
>
> 1. Input reviews (8 for Amazon, 10 for Oposum+ and Flipkart)
>
> 2. Aspects (Selected using aspect mapping)
>
> 3. Pseudo-summary (Selected through our synthetic dataset creation strategy)
>
> ### E. Clarification on Faithfulness
>
> Although Line 11 and Line 261 do mention that the faithfulness evaluation is done for general opinion summarization, we apologize for missing the inclusion of this vital information in some key places. (Will be added in the paper)
>
> Rebuttal Table 5 gives examples showing the problem of faithfulness in Multimodalsum generated summaries (Will be added to the paper). In comparison, we observe that our model-generated summaries tend to be faithful to the input in most cases. This we quantitatively measure using the different faithfulness evaluation metrics and human evaluations (Tables 5 and 6, Rebuttal Table 4).
>
> As mentioned in line 49, we specifically propose to curate synthetic datasets in such a way that it enables models to generate summaries that are more faithful to the input. Through experimentation, we show that our synthetic dataset creation strategy works better in generating faithful summaries compared to others. Line 769 mentions the reason for choosing SummaC, CTC, and FactCC. These metrics correlate better with human judgments for evaluating faithfulness [2]. Additionally, we show results for FactGraph in Rebuttal Table 4. Better results on these metrics do indicate that our model-generated summaries are more faithful, thereby indicating that our synthetic dataset creation strategy does aid in training the model to generate faithful summaries.
>
> Additionally, we employ three Master’s students (will be added to the paper) to evaluate summaries on different criteria such as faithfulness, coherence, conciseness, and fluency. We observe that in human evaluations as well our models score higher in faithfulness thus proving our hypothesis.
>
>
> ### F. Reason why our models are performing better than baselines
> **1. General Opinion Summarization:**
>
> In opinion summarization, synthetic dataset curation is important due to the absence of any large-scale supervised datasets (reviews-summary pairs). Such synthetic datasets enable supervised training for models.
>
> In Appendix A, we discussed how existing approaches create synthetic datasets followed by the generation of opinion summaries. We, however, hypothesize that a set of reviews and a pseudo-summary can act as a potential pair only if the pseudo-summary is lexically (rouge 1) and semantically (cosine similarity) similar to the review set. We use the matrix construction approach to find such pairs which ensures that only highly relevant pairs are selected in our synthetic dataset. The simple intuition behind this is that during training we want to show the model a pseudo-summary that is most relevant to the input reviews. This will enable the model to learn the task of opinion summarization much more accurately. This is evidenced by results in Table 4 as well as faithfulness scores in Tables 5 and 6 in the paper.
>
> **2. Aspect-specific Opinion Summarization:**
>
> Existing approaches are dependent on human-specified (pre-specified) and seed words for creating synthetic datasets for training and filtering sentences from reviews before generating summaries for inferencing.
>
> Our approach, however, creates aspect mappings automatically which enables the creation of synthetic datasets for a wide range of aspects. Here as well we use the combination of lexical and semantic scores for selecting a pseudo-summary (set of sentences about an aspect) and reviews pair. This ensures that the most relevant review-pseudo summary pair is shown to the model during training enabling better learning for the model to generate summaries. Our synthetic datasets are not restricted by the human-specified aspects and seed words which makes the model more robust in generating summaries for any relevant aspect of a specific product. This we intuit is causing improved scores of our models over existing models even though existing models have this extra aid of human-specified aspects and seed words for training and inference.
>
> ### G. Clustering Process
>
> We used the fast clustering library provided by SBERT [7]. We did not find the specific details of the algorithm. Specific details regarding its implementation will be added to the Appendix.
>
> ### H. Synthetic Triplets Creation
> Thank you for your comments. We will reframe this section to make the process easier to understand, with cleaner, understandable notations.
>
> ### H. Relevant related work in Appendix
> We do agree that some of the relevant work is in the appendix but it was primarily due to space limitations (4 pages for short paper). We will try to accommodate it in the main paper.
>
> ### I. Grammar, Presentation, and Missing References
> Thank you for pointing out the issues. Will incorporate all the suggestions in the paper.
>
> ### J. References
> [1] [Convex Aggregation for Opinion Summarization](https://aclanthology.org/2021.findings-emnlp.328/) (Iso et al., Findings 2021)
>
> [2] [X-FACTOR: A Cross-metric Evaluation of Factual Correctness in Abstractive Summarization](https://aclanthology.org/2022.emnlp-main.478/) (Chaudhury et al., 2022)
>
> [3] [Aspect-Controllable Opinion Summarization](https://aclanthology.org/2021.emnlp-main.528/) (Amplayo et al., 2021)
>
> [4] [Simple Yet Effective Synthetic Dataset Construction for Unsupervised Opinion Summarization](https://aclanthology.org/2023.findings-eacl.142/) (Shen et al., 2023)
>
> [5] [Ups and Downs: Modeling the Visual Evolution of Fashion Trends with One-Class Collaborative Filtering](https://arxiv.org/abs/1602.01585) (He and McAuley, 2016)
>
> [6] [FactGraph: Evaluating Factuality in Summarization with Semantic Graph Representations](https://aclanthology.org/2022.naacl-main.236/) (Ribeiro et al., NAACL 2022)
>
> [7] [Sentence-BERT: Sentence Embeddings using Siamese BERT-Networks](https://aclanthology.org/D19-1410/) (Reimers & Gurevych, EMNLP-IJCNLP 2019)

---

### Official Review · Reviewer_7Tmj · 2023-08-02

**Soundness:** 4

**Excitement:**

3: Ambivalent: It has merits (e.g., it reports state-of-the-art results, the idea is nice), but there are key weaknesses (e.g., it describes incremental work), and it can significantly benefit from another round of revision. However, I won't object to accepting it if my co-reviewers champion it.

**Missing References:**

The list of above:
* [1] [Convex Aggregation for Opinion Summarization](https://aclanthology.org/2021.findings-emnlp.328/) (Iso et al., Findings 2021)
* [2] [Prompted Opinion Summarization with GPT-3.5](https://aclanthology.org/2023.findings-acl.591/) (Bhaskar et al., Findings 2023)
* [3] [Attributable and Scalable Opinion Summarization](https://aclanthology.org/2023.acl-long.473/) (Hosking et al., ACL 2023)
* [4] [MeanSum: A Neural Model for Unsupervised Multi-document Abstractive Summarization](http://proceedings.mlr.press/v97/chu19b/chu19b.pdf) (Chu et al., ICML 2019)
* [5] [Extractive Opinion Summarization in Quantized Transformer Spaces](https://aclanthology.org/2021.tacl-1.17/) (Angelidis et al., TACL 2021)
* [6] [Self-Supervised and Controlled Multi-Document Opinion Summarization](https://aclanthology.org/2021.eacl-main.141) (Elsahar et al., EACL 2021)

**Paper Topic And Main Contributions:**

This study focuses on building a synthetic training dataset for opinion summarization. Creating such a dataset is crucial to developing a high-quality opinion summarization system without relying on a supervised training dataset.

The main contribution is the proposal of methods not only for general opinion summarization but also for aspect-based opinion summarization.

**Questions For The Authors:**

- Is there a specific reason for focusing on the e-commerce domain? Other well-studied opinions summarization benchmarks, such as Yelp [4] and Space [5], could also be considered. Could you clarify the reason for this focus?
- The synthetic dataset creation method bears similarities to an existing study [6]. While differences exist, it would be nice to acknowledge their work.

* [4] [MeanSum: A Neural Model for Unsupervised Multi-document Abstractive Summarization](http://proceedings.mlr.press/v97/chu19b/chu19b.pdf) (Chu et al., ICML 2019)
* [5] [Extractive Opinion Summarization in Quantized Transformer Spaces](https://aclanthology.org/2021.tacl-1.17/) (Angelidis et al., TACL 2021)
* [6] [Self-Supervised and Controlled Multi-Document Opinion Summarization](https://aclanthology.org/2021.eacl-main.141) (Elsahar et al., EACL 2021)




**Reasons To Accept:**

- The proposed method demonstrates improvements across two different tasks: aspect-based and general opinion summarizations, outperforming strong baselines.
- The authors conducted a human evaluation using best-worst scaling, a credible approach, which showed that the proposed method performs better or comparably to the baselines.
- Overall, the experiments look solid and easily reproducible.


**Reasons To Reject:**

- The proposed method introduces too many hyper-parameters, potentially limiting its generalizability to other domains.
- An important baseline, Coop [1], for unsupervised general opinion summarization, which exhibits better performance than the author's model in terms of ROUGE-1 & L on Amazon, is missing.
- The discussion of Language Model-based methods (LLMs) is lacking. Given the availability of data for training, it would be relevant to consider studies on opinion summarization with LLMs [2], [3]. It would be beneficial to differentiate why synthetic data creation remains relevant in the post-ChatGPT era.

* [1] [Convex Aggregation for Opinion Summarization](https://aclanthology.org/2021.findings-emnlp.328/) (Iso et al., Findings 2021)
* [2] [Prompted Opinion Summarization with GPT-3.5](https://aclanthology.org/2023.findings-acl.591/) (Bhaskar et al., Findings 2023)
* [3] [Attributable and Scalable Opinion Summarization](https://aclanthology.org/2023.acl-long.473/) (Hosking et al., ACL 2023)

**Reproducibility:**

4: Could mostly reproduce the results, but there may be some variation because of sample variance or minor variations in their interpretation of the protocol or method.

**Reviewer Confidence:**

4: Quite sure. I tried to check the important points carefully. It's unlikely, though conceivable, that I missed something that should affect my ratings.

---

> ### Author Rebuttal · Authors · 2023-08-29
>
> We thank the reviewer for the valuable suggestions and feedback. We have tried our best to resolve all the doubts and provide clarifications. We request the reviewer to kindly consider increasing the scores positively if they find the rebuttal satisfactory.
>
> ### A. Additional Results
>
> **1. General Opinion Summarization:** Due to certain issues with our code, our results in the paper are with beam size 1. We re-ran our updated code with different beam sizes and observed the best performance using beam size 5 (decided using the dev set). Currently, we achieve the best R2 and RL score and second-best R1 compared to baselines.
>
> | *Amazon* | R1 | R2 | RL |
> | ------------- | ---- | ---- | ---- |
> | Multimodalsum | 34.19 | 7.05 | 20.81 |
> | ConsistSum | 33.32 | 5.94 | _21.41_ |
> | TransSum | 34.23 | _7.24_ | 20.49 |
> | COOP [1] | **36.57** | 7.23 | 21.24 |
> | Ours | _35.46*_ | **7.30** | **21.50** |
>
> **Rebuttal Table 1:** **Bold** and _italics_ indicate best and second-best results. *indicates results that are significantly better than MultimodalSum with a pvalue<0.05 on paired t-test computed over the Amazon test set for general opinion summarization (couldn't be compared with others due to unavailability of code/results).
>
>
> **2. Aspect-specific Opinion Summarization:** Due to certain issues with our code, our results in the paper are with beam size 1. We re-ran our updated code with different beam sizes and observed the best performance using beam size 2 (decided using the dev set). Currently, we achieve the best R1, R2, and RL scores compared to baselines.
>
> | *Oposum+* | R1 | R2 | RL |
> | ------------- | ---- | ---- | ---- |
> | Acesum | 29.53 | 6.79 | _21.06_ |
> | SW-LOO | _30.00_ | **6.92** | 20.76 |
> | NLI-LOO | 28.90 | 6.60 | 20.11 |
> | Ours | **30.95*** | **6.92** | **21.73** |
>
> **Rebuttal Table 2:** **Bold** and _italics_ indicate best and second-best results. *indicates results that are significantly better than AceSum with a pvalue<0.05 on paired t-test computed over the Oposum+ test set for aspect-specific opinion summarization (couldn't be compared with others due to unavailability of code/results).
>
>
> | Acesum |  |
> | --------- | ---------------- |
> | Ports | the hdmi ports stopped working and the tv would go black for about three to five seconds every few minutes. it has all of the inputs for us, and more. |
> | Picture | the picture quality was better than my 5 year old lcd samsung. the picture is decent, but not as good as the htpc.|
> | Sound | i bought the tv at fry's after salesperson assured me headphones would work with this. picture is acceptable and sound is above average for the price. |
> | **Ours** |  |
> | Ports | i have had this tv for a few months now and i have had no issues with it, it has all of the hdmi ports i need and the picture quality is great. |
> | Picture | the picture is great, the tv is easy to set up, and it has all of the inputs i need, but it is not a smart tv. |
> | Sound | the sound is decent, but i have a surround sound system so i don't use the tv's speakers much, but it is a great tv for the price.|
>
> **Rebuttal Table 3:** Comparative analysis of our model-generated summaries with Acesum for aspect-specific opinion summarization on different aspects such as _ports, picture,_ and _sound_. Our model does not rely on any human-specified aspects and seed words for training, making our approach generalizable to other domains where such aspects and seed words are not defined.
>
> **3. Faithfulness Evaluation:**  We additionally show results of general opinion summarization on FactGraph [3] which is a metric that correlates the best with human judgment on faithfulness evaluation [2].
>
> |  | FactGraph |
> | ------------- | ---- |
> | Plansum | 0.16 |
> | Multimodalsum | 0.53 |
> | Ours | **0.66** |
>
> **Rebuttal Table 4:** **Bold** indicate the best result. We achieved the highest score on FactGraph for faithfulness evaluation for general opinion summarization.
>
> | Models | Summaries |
> | --------- | ---------------- |
> | Multimodalsum | This is a good receiver for the price. I have it **hooked up to a pair of Sony speakers** and it sounds great. The only problem I have with it is that the **remote control is a little difficult to use**. I would recommend this to a friend. |
> | Ours | This is a great stereo for the price. The only thing I don't like about it is the fact that you can't adjust the volume on the radio. I have to turn it up all the way to hear it. Other than that, it's a great product. |
> | Multimodalsum | This is a nice teapot, but the color is not as bright as the picture. It is more of a dark turquoise than a light blue. **I was hoping it would be more of an aqua blue, but it is more like a dark aqua**. It still looks nice, but I would have preferred the color to be more like the photo. |
> | Ours | This is a beautiful teapot, but the color is not the same as shown in the picture. It is more of a dark turquoise. The color is a little darker than the picture, but it is still a beautiful color. The only thing I don't like about it is that the handle is very hard to open.|
>
> **Rebuttal Table 5:** Example of faithfulness issue in baseline models compared to ours. There is no support for the information in **bold** parts in the corresponding input.
>
> ### B. Clarification on hyper-parameters
> We do agree that there are some extra hyper-parameters that would change according to the domain under consideration. However, having these hyper-parameters is essential for the proper selection of review-pseudo summary pairs to fine-tune models.
>
> ### C. COOP baseline
> We apologize for missing this out. Our current results in Rebuttal Table 1 do show that our model surpasses Coop on both R2 and RL. Thanks for pointing this out.
>
> ### D. LLM discussion
> We couldn't compare our models with [3], and [4] as they are very recent papers. As mentioned in the limitations section, we were investigating better synthetic dataset approaches and used BART to have a fair comparison with previous models. We do agree the applicability of these for LLMs needs to be investigated.
>
> ### E. Reason for focus on e-commerce datasets
> The main aim of the paper was to show the applicability of our approach not restricted to just one e-commerce dataset but others as well. We specifically chose e-commerce as there were multiple datasets available. This ensured that the better results were not observed only on a specific dataset but on the whole e-commerce domain. Appendix B does talk about the same approach being applicable to other domains but this needs to be demonstrated empirically.
>
> ### E. Missing References
> Thank you for pointing out the missing references. Will incorporate them in the paper.
>
> ### F. References
> [1] [Convex Aggregation for Opinion Summarization](https://aclanthology.org/2021.findings-emnlp.328/) (Iso et al., Findings 2021)
>
> [2] [X-FACTOR: A Cross-metric Evaluation of Factual Correctness in Abstractive Summarization](https://aclanthology.org/2022.emnlp-main.478/) (Chaudhury et al., EMNLP 2022)
>
> [3] [FactGraph: Evaluating Factuality in Summarization with Semantic Graph Representations](https://aclanthology.org/2022.naacl-main.236/) (Ribeiro et al., NAACL 2022)
>
> [4] [Prompted Opinion Summarization with GPT-3.5](https://aclanthology.org/2023.findings-acl.591/) (Bhaskar et al., Findings 2023)
>
> [5] [Attributable and Scalable Opinion Summarization](https://aclanthology.org/2023.acl-long.473/) (Hosking et al., ACL 2023)

---

### Official Review · Reviewer_UMd3 · 2023-08-06

**Typos Grammar Style And Presentation Improvements:** Line 35 can contain 2-3 recent papers…
**Soundness:** 3

**Excitement:**

3: Ambivalent: It has merits (e.g., it reports state-of-the-art results, the idea is nice), but there are key weaknesses (e.g., it describes incremental work), and it can significantly benefit from another round of revision. However, I won't object to accepting it if my co-reviewers champion it.

**Paper Topic And Main Contributions:**

In this study, the researchers address the limitations of aspect-specific summarization, which currently relies heavily on human-specified aspects and seed words. Additionally, general summarization struggles to accurately capture the essence of input reviews. To overcome these challenges, the authors propose innovative techniques for generating synthetic datasets specifically tailored for both aspect-specific and general summarization. By utilizing cosine similarity and ROUGE-1 scores for pseudo-summary selection, they achieve notable improvements of 0.02 and 0.2 points on the SummaC and FactCC benchmarks, respectively. This approach represents a  step forward compared to previous methods that heavily depended on human-specified aspects and aspect-specific seed words for generating data for aspect-specific summarization.

**Questions For The Authors:**

1. Clustering Process:
The paper lacks specific details regarding the clustering algorithm employed for both fine-grained and coarse clusters. Including a description of the clustering algorithms utilized would enhance the readers' understanding of the methods employed to group data points into meaningful clusters.
2. Intersection of Coarse-Grained and Fine-Grained Aspects:
The paper briefly mentions an intersection between coarse-grained and fine-grained aspects at line 110. However, it does not elaborate on how this intersection is measured. Clarifying whether it is based on word overlap or some other metric would provide clarity and reinforce the validity of the findings.
3. Purpose of Paragraph at Line 117:
The purpose of creating these triplets in the paragraph starting at line 117 requires further explanation. Providing an introduction or overview of the intention behind this approach before delving into the technical details would be beneficial, as it would set the context and enable readers to grasp the significance of the triplet creation process more effectively.
4. Definition of the term D:
At line 138, the paper references the term "D" without providing a clear definition. To ensure clarity and avoid confusion, it is crucial to explicitly define "D" within the paper, preferably in the section where it is first introduced. This would enable readers to comprehend the role and significance of "D" in the context of the study.

**Reasons To Accept:**

In this research endeavor, the authors harness the power of recent advances in machine learning models to generate aspects. They showcase their ingenuity by developing clever methodologies for creating pseudo summaries. By leveraging cutting-edge techniques, they aim to improve the quality and accuracy of aspects and summaries in their work.

**Reasons To Reject:**

Analysis of results - The results section provides a verbose representation of the result tables. However, it lacks an in-depth analysis of why certain results are as they are. To enhance the section, it would be beneficial to include a detailed analysis explaining the underlying reasons behind the observed outcomes. This would provide valuable insights and a better understanding of the significance and implications of the results obtained.

Comparisons to baselines that uses human specified aspects? - In comparison to a baseline that relies on human-specified aspects (AceSum), the authors have not included any comparative analysis in their paper with respect to their closest competitor model concerning the number of added data points. The paper lacks a discussion on the usefulness of automatically generating these aspects using InstructABSA. Consequently, certain pertinent questions remain unanswered, potentially limiting a comprehensive understanding of the research findings. A more thorough investigation into these aspects would significantly strengthen the paper's contributions and provide valuable insights for the readers.

**Reproducibility:**

1: Could not reproduce the results here no matter how hard they tried.

**Reviewer Confidence:**

4: Quite sure. I tried to check the important points carefully. It's unlikely, though conceivable, that I missed something that should affect my ratings.

---

> ### Author Rebuttal · Authors · 2023-08-29
>
> We thank the reviewer for the valuable suggestions and feedback. We have tried our best to resolve all the doubts and provide clarifications. We request the reviewer to kindly consider increasing the scores positively if they find the rebuttal satisfactory.
>
> ### A. Additional Results
>
> **1. General Opinion Summarization:** Due to certain issues with our code, our results in the paper are with beam size 1. We re-ran our updated code with different beam sizes and observed the best performance using beam size 5 (decided using the dev set). Currently, we achieve the best R2 and RL score and second-best R1 compared to baselines.
>
> | *Amazon* | R1 | R2 | RL |
> | ------------- | ---- | ---- | ---- |
> | Multimodalsum | 34.19 | 7.05 | 20.81 |
> | ConsistSum | 33.32 | 5.94 | _21.41_ |
> | TransSum | 34.23 | _7.24_ | 20.49 |
> | COOP [1] | **36.57** | 7.23 | 21.24 |
> | Ours | _35.46*_ | **7.30** | **21.50** |
>
> **Rebuttal Table 1:** **Bold** and _italics_ indicate best and second-best results. *indicates results that are significantly better than MultimodalSum with a pvalue<0.05 on paired t-test computed over the Amazon test set for general opinion summarization (couldn't be compared with others due to unavailability of code/results).
>
>
> **2. Aspect-specific Opinion Summarization:** Due to certain issues with our code, our results in the paper are with beam size 1. We re-ran our updated code with different beam sizes and observed the best performance using beam size 2 (decided using the dev set). Currently, we achieve the best R1, R2, and RL scores compared to baselines.
>
> | *Oposum+* | R1 | R2 | RL |
> | ------------- | ---- | ---- | ---- |
> | Acesum | 29.53 | 6.79 | _21.06_ |
> | SW-LOO | _30.00_ | **6.92** | 20.76 |
> | NLI-LOO | 28.90 | 6.60 | 20.11 |
> | Ours | **30.95*** | **6.92** | **21.73** |
>
> **Rebuttal Table 2:** **Bold** and _italics_ indicate best and second-best results. *indicates results that are significantly better than AceSum with a pvalue<0.05 on paired t-test computed over the Oposum+ test set for aspect-specific opinion summarization (couldn't be compared with others due to unavailability of code/results).
>
>
> | Acesum |  |
> | --------- | ---------------- |
> | Ports | the hdmi ports stopped working and the tv would go black for about three to five seconds every few minutes. it has all of the inputs for us, and more. |
> | Picture | the picture quality was better than my 5 year old lcd samsung. the picture is decent, but not as good as the htpc.|
> | Sound | i bought the tv at fry's after salesperson assured me headphones would work with this. picture is acceptable and sound is above average for the price. |
> | **Ours** |  |
> | Ports | i have had this tv for a few months now and i have had no issues with it, it has all of the hdmi ports i need and the picture quality is great. |
> | Picture | the picture is great, the tv is easy to set up, and it has all of the inputs i need, but it is not a smart tv. |
> | Sound | the sound is decent, but i have a surround sound system so i don't use the tv's speakers much, but it is a great tv for the price.|
>
> **Rebuttal Table 3:** Comparative analysis of our model-generated summaries with Acesum for aspect-specific opinion summarization on different aspects such as _ports, picture,_ and _sound_. Our model does not rely on any human-specified aspects and seed words for training, making our approach generalizable to other domains where such aspects and seed words are not defined.
>
> **3. Faithfulness Evaluation:**  We additionally show results of general opinion summarization on FactGraph [3] which is a metric that correlates the best with human judgment on faithfulness evaluation [2].
>
> |  | FactGraph |
> | ------------- | ---- |
> | Plansum | 0.16 |
> | Multimodalsum | 0.53 |
> | Ours | **0.66** |
>
> **Rebuttal Table 4:** **Bold** indicate the best result. We achieved the highest score on FactGraph for faithfulness evaluation for general opinion summarization.
>
> | Models | Summaries |
> | --------- | ---------------- |
> | Multimodalsum | This is a good receiver for the price. I have it **hooked up to a pair of Sony speakers** and it sounds great. The only problem I have with it is that the **remote control is a little difficult to use**. I would recommend this to a friend. |
> | Ours | This is a great stereo for the price. The only thing I don't like about it is the fact that you can't adjust the volume on the radio. I have to turn it up all the way to hear it. Other than that, it's a great product. |
> | Multimodalsum | This is a nice teapot, but the color is not as bright as the picture. It is more of a dark turquoise than a light blue. **I was hoping it would be more of an aqua blue, but it is more like a dark aqua**. It still looks nice, but I would have preferred the color to be more like the photo. |
> | Ours | This is a beautiful teapot, but the color is not the same as shown in the picture. It is more of a dark turquoise. The color is a little darker than the picture, but it is still a beautiful color. The only thing I don't like about it is that the handle is very hard to open.|
>
> **Rebuttal Table 5:** Example of faithfulness issue in baseline models compared to ours. There is no support for the information in **bold** parts in the corresponding input.
>
>
> ### B. Reason why our models are performing better than baselines
> **1. General Opinion Summarization:**
>
> In opinion summarization, synthetic dataset curation is important due to the absence of any large-scale supervised datasets (reviews-summary pairs). Such synthetic datasets enable supervised training for models.
>
> In Appendix A, we discussed how existing approaches create synthetic datasets followed by the generation of opinion summaries. We, however, hypothesize that a set of reviews and a pseudo-summary can act as a potential pair only if the pseudo-summary is lexically (rouge 1) and semantically (cosine similarity) similar to the review set. We use the matrix construction approach to find such pairs which ensures that only highly relevant pairs are selected in our synthetic dataset. The simple intuition behind this is that during training we want to show the model a pseudo-summary that is most relevant to the input reviews. This will enable the model to learn the task of opinion summarization much more accurately. This is evidenced by results in Table 4 as well as faithfulness scores in Tables 5 and 6 in the paper.
>
> **2. Aspect-specific Opinion Summarization:**
>
> Existing approaches are dependent on human-specified (pre-specified) and seed words for creating synthetic datasets for training and filtering sentences from reviews before generating summaries for inferencing.
>
> Our approach, however, creates aspect mappings automatically which enables the creation of synthetic datasets for a wide range of aspects. Here as well we use the combination of lexical and semantic scores for selecting a pseudo-summary (set of sentences about an aspect) and reviews pair. This ensures that the most relevant review-pseudo summary pair is shown to the model during training enabling better learning for the model to generate summaries. Our synthetic datasets are not restricted by the human-specified aspects and seed words which makes the model more robust in generating summaries for any relevant aspect of a specific product. This we intuit is causing improved scores of our models over existing models even though existing models have this extra aid of human-specified aspects and seed words for training and inference.
>
> ### C. Clustering Process
>
> We used the fast clustering library provided by SBERT [4]. We did not find the specific details of the algorithm. Specific details regarding its implementation will be added to the Appendix.
>
> ### D. Clarification on coarse-grained and fine-grained aspects
> The intersection is on the aspect level. We will add this specific detail in the paper.
>
>
> ### E. Comment on the suggestions
> 1. Purpose of Paragraph at Line 117: Thanks for the suggestion. Will add appropriate reasoning at the start of the section.
>
> 2. Definition of the term D: We apologize for this mistake and will update it in the paper. We had made some last-minute section shifting which caused this lapse.
>
> 3. Presentation improvements: Noted. Will be updated in the paper.
>
> ### F. References
> [1] [Convex Aggregation for Opinion Summarization](https://aclanthology.org/2021.findings-emnlp.328/) (Iso et al., Findings 2021)
>
> [2] [X-FACTOR: A Cross-metric Evaluation of Factual Correctness in Abstractive Summarization](https://aclanthology.org/2022.emnlp-main.478/) (Chaudhury et al., EMNLP 2022)
>
> [3] [FactGraph: Evaluating Factuality in Summarization with Semantic Graph Representations](https://aclanthology.org/2022.naacl-main.236/) (Ribeiro et al., NAACL 2022)
>
> [4] [Sentence-BERT: Sentence Embeddings using Siamese BERT-Networks](https://aclanthology.org/D19-1410/) (Reimers & Gurevych, EMNLP-IJCNLP 2019)

---

### Meta-Review · Area_Chair_ZHpW · 2023-09-17

**Recommendation:** 4

**Metareview:**

The paper introduces methods for creating synthetic training data to improve opinion summarization, addressing both general and aspect-based summarization. The paper demonstrates improvements over strong baselines in various evaluation metrics, supported by human assessment. However, concerns were raised in the reviews about the introduction of numerous hyperparameters, potential limitations in generalizability, the absence of a baseline comparison with Coop in unsupervised general opinion summarization, and the need for a more comprehensive discussion of language model-based methods in the post-ChatGPT era. Additionally, issues related to presentation clarity, ambiguities, and details in various sections were noted, along with recommendations for a fairer comparison and a clearer motivation regarding faithfulness issues in other synthetic datasets. Despite these issues, reviewers were satisfied by most of the responses of the authors during the rebuttal period.

---

### Decision · Program_Chairs · 2023-10-07

**Decision:**

Accept-Findings

**Comment:**

The paper introduces methods for creating synthetic training data to improve opinion summarization, addressing both general and aspect-based summarization. The paper demonstrates improvements over strong baselines in various evaluation metrics, supported by human assessment. However, concerns were raised in the reviews about the introduction of numerous hyperparameters, potential limitations in generalizability, the absence of a baseline comparison with Coop in unsupervised general opinion summarization, and the need for a more comprehensive discussion of language model-based methods in the post-ChatGPT era. Additionally, issues related to presentation clarity, ambiguities, and details in various sections were noted, along with recommendations for a fairer comparison and a clearer motivation regarding faithfulness issues in other synthetic datasets. Despite these issues, reviewers were satisfied by most of the responses of the authors during the rebuttal period.